# Neural Graph Evolution: Towards Efficient Automatic Robot Design

**Tingwu Wang**[1,2*]**, Yuhao Zhou**[1,2*]**, Sanja Fidler**[1,2,3] **& Jimmy Ba**[1,2]
[1] Department of Computer Science, University of Toronto
[2] Vector Institute
[3] NVIDIA
`{tingwuwang,henryzhou,fidler,jba}@cs.toronto.edu`

## Abstract

Despite the recent successes in robotic locomotion control, the design of robots, i.e., the design of their body structure, still heavily relies on human engineering. Automatic robot design has been a long studied subject, however, progress has been slow due to large combinatorial search space and the difficulty to efficiently evaluate the candidate structures. Note that one needs to both, search over many possible body structures, and choose among them based on how the robot with that structure performs in an environment. The latter means training an optimal controller given a candidate structure, which in itself is costly to obtain. In this paper, we propose Neural Graph Evolution (NGE), which performs evolutionary search in graph space, by iteratively evolving graph structures using simple mutation primitives. Key to our approach is to parameterize the control policies with graph neural networks, which allows us to transfer skills from previously evaluated designs during the graph search. This significantly reduces evaluation cost of new candidates and makes the search process orders of magnitude more efficient than that of past work. In addition, NGE applies Graph Mutation with Uncertainty (GM-UC) by incorporating model uncertainty, which reduces the search space by balancing exploration and exploitation. We show that NGE significantly outperforms previous methods in terms of convergence rate and final performance. As shown in experiments, NGE is the first algorithm that can automatically discover kinematically preferred robotic graph structures, such as a fish with two symmetric flat side-fins and a tail, or a cheetah with athletic front and back legs. NGE is extremely efficient, it finds plausible robotic structures within a day on a single 64 CPU-core Amazon EC2 machine.

## 1 Introduction

The goal of robot design is to find an optimal body structure and its means of locomotion to best achieve a given objective in an environment. Robot design often relies on careful human-engineering and expert knowledge. The field of automatic robot design aims to search for these structures automatically. This has been a long-studied subject, however, with limited success. There are two major challenges: 1) the search space of all possible designs is large and combinatorial, and 2) the evaluation of each design requires learning or testing a separate optimal controller that is often expensive to obtain.

In (Sims, 1994), the authors evolved creatures with 3D-blocks. Recently, soft robots have been studied in (Joachimczak et al., 2014), which were evolved by adding small cells connected to the old ones. In (Cheney et al., 2014), the 3D voxels were treated as the minimum element of the robot. Most evolutionary robots (Duff et al., 2001; Neri, 2010) require heavy engineering of the initial structures, evolving rules and careful human-guidance. Due to the combinatorial nature of the problem, evolutionary, genetic or random structure search have been the de facto algorithms of automatic robot design in the pioneering works (Sims, 1994; Steels, 1993; Mitchell & Forrest, 1994; Langton, 1997; Lee, 1998; Taylor, 2017; Calandra et al., 2016). In terms of the underlying algorithm,

---

*Two authors contribute equally.

most of these works have a similar population-based optimization loop to the one used in (Sims, 1994). None of these algorithms are able to evolve kinematically reasonable structures, as a result of large search space and the inefficient evaluation of candidates.

Similar in vein to automatic robot design, automatic neural architecture search also faces a large combinatorial search space and difficulty in evaluation. There have been several approaches to tackle these problems. Bayesian optimization approaches (Snoek et al., 2012) primarily focus on fine-tuning the number of hidden units and layers from a predefined set. Reinforcement learning (Zoph & Le, 2016) and genetic algorithms (Liu et al., 2017) are studied to evolve recurrent neural networks (RNNs) and convolutional neural networks (CNNs) from scratch in order to maximize the validation accuracy. These approaches are computationally expensive because a large number of candidate networks have to be trained from grounds up. (Pham et al., 2018) and (Stanley & Miikkulainen, 2002) propose weight sharing among all possible candidates in the search space to effectively amortize the inner loop training time and thus speed up the architecture search. A typical neural architecture search on ImageNet (Krizhevsky et al., 2012) takes 1.5 days using 200 GPUs (Liu et al., 2017).

In this paper, we propose an efficient search method for automatic robot design, *Neural Graph Evolution* (NGE), that co-evolves both, the robot design and the control policy. Unlike the recent reinforcement learning work, where the control policies are learnt on specific robots carefully designed by human experts (Mnih et al., 2013; Bansal et al., 2017; Heess et al., 2017), NGE aims to adapt the robot design along with policy learning to maximize the agent's performance. NGE formulates automatic robot design as a graph search problem. It uses a graph as the main backbone of rich design representation and graph neural networks (GNN) as the controller. This is key in order to achieve efficiency of candidate structure evaluation during evolutionary graph search. Similar to previous algorithms like (Sims, 1994), NGE iteratively evolves new graphs and removes graphs based on the performance guided by the learnt GNN controller. The specific contributions of this paper are as follows:

- We formulate the automatic robot design as a graph search problem.
- We utilize graph neural networks (GNNs) to share the weights between the controllers, which greatly reduces the computation time needed to evaluate each new robot design.
- To balance exploration and exploitation during the search, we developed a mutation scheme that incorporates model uncertainty of the graphs.

We show that NGE automatically discovers robot designs that are comparable to the ones designed by human experts in MuJoCo (Todorov et al., 2012), while random graph search or naive evolutionary structure search (Sims, 1994) fail to discover meaningful results on these tasks.

## 2 BACKGROUND

### 2.1 REINFORCEMENT LEARNING

In reinforcement learning (RL), the problem is usually formulated as a Markov Decision Process (MDP). The infinite-horizon discounted MDP consists of a tuple of $(\mathcal{S}, \mathcal{A}, \gamma, P, R)$, respectively the state space, action space, discount factor, transition function, and reward function. The objective of the agent is to maximize the total expected reward $J(\theta) = \mathbb{E}_\pi \left[ \sum_{t=0}^{\infty} \gamma^t r(s_t, a_t) \right]$, where the state transition follows the distribution $P(s_{t+1}|s_t, a_t)$. Here, $s_t$ and $a_t$ denotes the state and action at time step $t$, and $r(s_t, a_t)$ is the reward function. In this paper, to evaluate each robot structure, we use PPO to train RL agents (Schulman et al., 2017; Heess et al., 2017). PPO uses a neural network parameterized as $\pi_\theta(a_t|s_t)$ to represent the policy, and adds a penalty for the KL-divergence between the new and old policy to prevent over-optimistic updates. PPO optimizes the following surrogate objective function instead:

$$J_{\text{PPO}}(\theta) = \mathbb{E}_{\pi_\theta} \left[ \sum_{t=0}^{\infty} A^t(s_t, a_t) r^t(s_t, a_t) \right] - \beta \, \text{KL} \left[ \pi_\theta(: |s_t) | \pi_{\theta_{old}}(: |s_t) \right]. \tag{1}$$

We denote the estimate of the expected total reward given the current state-action pair, the value and the advantage functions, as $Q^t(s_t, a_t)$, $V(s_t)$ and $A^t(s_t, a_t)$ respectively. PPO solves the problem by iteratively generating samples and optimizing $J_{\text{PPO}}$ (Schulman et al., 2017).

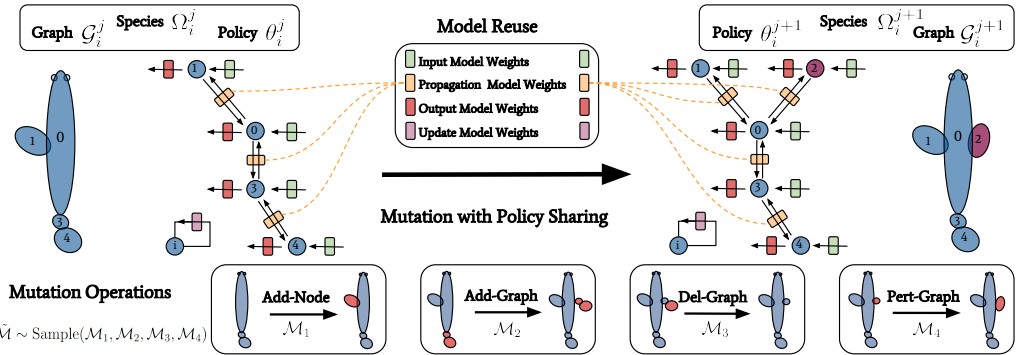

Figure 1: In NGE, several mutation operations are allowed. By using Policy Sharing, child species reuse weights from parents, even if the graphs are different. The same color indicates shared and reused weights. For better visualization, we only plot the sharing of propagation model (yellow curves).

## 2.2 GRAPH NEURAL NETWORK

Graph Neural Networks (GNNs) are suitable for processing data in the form of graph (Bruna et al., 2014; Defferrard et al., 2016; Li et al., 2015; Kipf & Welling, 2017; Duvenaud et al., 2015; Henaff et al., 2015). Recently, the use of GNNs in locomotion control has greatly increased the transferability of controllers (Wang et al., 2018). A GNN operates on a graph whose nodes and edges are denoted respectively as $u \in V$ and $e \in E$. We consider the following GNN, where at timestep $t$ each node in GNN receives an input feature and is supposed to produce an output at a node level.

**Input Model**: The input feature for node $u$ is denoted as $x_u^t$. $x_u^t$ is a vector of size $d$, where $d$ is the size of features. In most cases, $x_u^t$ is produced by the output of an embedding function used to encode information about $u$ into $d$-dimensional space.

**Propagation Model**: Within each timestep $t$, the GNN performs $\mathcal{T}$ internal propagations, so that each node has global (neighbourhood) information. In each propagation, every node communicates with its neighbours, and updates its hidden state by absorbing the input feature and message. We denote the hidden state at the internal propagation step $\tau$ ($\tau \leq \mathcal{T}$) as $h_u^{t;\tau}$. Note that $h_u^{t;0}$ is usually initialized as $h_u^{t-1,\mathcal{T}}$, i.e., the final hidden state in the previous time step. $h^{0,0}$ is usually initialized to zeros. The message that $u$ sends to its neighbors is computed as

$$m_u^{t,\tau} = M(h_u^{t,\tau-1}), \tag{2}$$

where $M$ is the message function. To compute the updated $h_u^{t,\tau}$, we use the following equations:

$$r_u^{t,\tau} = R(\{m_v^{t,\tau} \mid \forall v \in \mathcal{N}_G(u)\}), \; h_u^{t,\tau} = U(h_u^{t,\tau-1}, (r_u^{t,\tau}; x_u^t)) \tag{3}$$

where $R$ and $U$ are the message aggregation function and the update function respectively, and $\mathcal{N}_G(u)$ denotes the neighbors of $u$.

**Output Model**: Output function $F$ takes input the node's hidden states after the last internal propagation. The node-level output for node $u$ is therefore defined as $\mu_u^t = F(h_u^{t;\mathcal{T}})$.

Functions $M, R, U, F$ in GNNs can be trainable neural networks or linear functions. For details of GNN controllers, we refer readers to (Wang et al., 2018).

## 3 NEURAL GRAPH EVOLUTION

In robotics design, every component, including the robot arms, finger and foot, can be regarded as a node. The connections between the components can be represented as edges. In locomotion control, the robotic simulators like MuJoCo (Todorov et al., 2012) use an XML file to record the graph of the robot. As we can see, robot design is naturally represented by a graph. To better illustrate Neural Graph Evolution (NGE), we first introduce the terminology and summarize the algorithm.

**Graph and Species**. We use an undirected *graph* $\mathcal{G} = (V, E, A)$ to represent each robotic design. $V$ and $E$ are the collection of physical body nodes and edges in the graph, respectively. The mapping

---

**Algorithm 1** Neural Graph Evolution

---

1: Initialize generation $\mathcal{P}^0 \leftarrow \{(\theta_i^0, G_i^0)\}_{i=1}^{\mathcal{N}}$
2: **while** Evolving $j$th generation **do**                     ▷ Evolution outer loop
3:      **for** $i$th species $(\theta_i^j, \mathcal{G}_i^j) \in \mathcal{P}^j$ **do**             ▷ Species fitness inner loop
4:          $\theta_i^{j+1} \leftarrow \text{Update}(\theta_i^j)$                  ▷ Train policy network
5:          $\xi_i \leftarrow \xi(\theta_i^{j+1}, G_i^j)$                    ▷ Evaluate fitness
6:      **end for**
7:      $\mathcal{P}^{j+1} \leftarrow \mathcal{P}^j \setminus \{(\theta_k^j, \mathcal{G}_k^j) \in \mathcal{P}^j, \forall k \in \arg\min_{\mathcal{K}}(\{\xi_i\})\}.$      ▷ Remove worst $\mathcal{K}$ species
8:      $\hat{\mathcal{P}} \leftarrow \{(\hat{\theta}_h, \hat{\mathcal{G}}_h = \mathcal{M}(\mathcal{G}_{h,p})), \text{where } \mathcal{G}_{h,p} \sim \text{Uniform}(\mathcal{P}^{j+1})\}_{h=1}^{\mathcal{C}}$    ▷ Mutate from survivors
9:      $\mathcal{P}^{j+1} \leftarrow \mathcal{P}^{j+1} \cup \{(\hat{\theta}_k, \hat{\mathcal{G}}_k) \in \hat{\mathcal{P}}, \forall k \in \arg\max_{\mathcal{K}}(\{\xi_P(\hat{G}_h)\})\}.$      ▷ Pruning
10: **end while**

---

$A : V \rightarrow \Lambda$ maps the node $u \in V$ to its structural attributes $A(u) \in \Lambda$, where $\Lambda$ is the attributes space. For example, the fish in Figure 1 consists of a set of ellipsoid nodes, and vector $A(u)$ describes the configurations of each ellipsoid. The controller is a policy network parameterized by weights $\theta$. The tuple formed by the graph and the policy is defined as a *species*, denoted as $\Omega = (\mathcal{G}, \theta)$.

**Generation and Policy Sharing**. In the $j$-th iteration, NGE evaluates a pool of species called a *generation*, denoted as $P^j = \{(\mathcal{G}_i^j, \theta_i^j), \forall i = 1, 2, ..., \mathcal{N}\}$, where $\mathcal{N}$ is the size of the generation. In NGE, the search space includes not only the graph space, but also the weight or parameter space of the policy network. For better efficiency of NGE, we design a process called Policy Sharing (PS), where weights are reused from parent to child species. The details of PS is described in Section 3.4.

Our model can be summarized as follows. NGE performs population-based optimization by iterating among *mutation*, *evaluation* and *selection*. The objective and performance metric of NGE are introduced in Section 3.1. In NGE, we randomly initialize the generation with $\mathcal{N}$ species. For each generation, NGE trains each species and evaluates their fitness separately, the policy of which is described in Section 3.2. During the selection, we eliminate $\mathcal{K}$ species with the worst fitness. To mutate $\mathcal{K}$ new species from surviving species, we develop a novel mutation scheme called Graph Mutation with Uncertainty (GM-UC), described in Section 3.3, and efficiently inherit policies from the parent species by Policy Sharing, described in Section 3.4. Our method is outlined in Algorithm 1.

## 3.1 AMORTIZED FITNESS AND OBJECTIVE FUNCTION

Fitness represents the performance of a given $\mathcal{G}$ using the optimal controller parameterized with $\theta^*(\mathcal{G})$. However, $\theta^*(\mathcal{G})$ is impractical or impossible to obtain for the following reasons. First, each design is computationally expensive to evaluate. To evaluate one graph, the controller needs to be trained and tested. Model-free (MF) algorithms could take more than one million in-game timesteps to train a simple 6-degree-of-freedom cheetah (Schulman et al., 2017), while model-based (MB) controllers usually require much more execution time, without the guarantee of having higher performance than MF controllers (Tassa et al., 2012; Nagabandi et al., 2017; Drews et al., 2017; Chua et al., 2018). Second, the search in robotic graph space can easily get stuck in local-optima. In robotic design, local-optima are difficult to detect as it is hard to tell whether the controller has converged or has reached a temporary optimization plateau. Learning the controllers is a computation bottleneck in optimization.

In population-based robot graph search, spending more computation resources on evaluating each species means that fewer different species can be explored. In our work, we enable transferablity between *different* topologies of NGE (described in Section 3.2 and 3.4). This allows us to introduce *amortized fitness* (AF) as the objective function across generations for NGE. AF is defined in the following equation as,

$$\xi(\mathcal{G}, \theta) = \mathbb{E}_{\pi_\theta, \mathcal{G}} \left[ \sum_{t=0}^{\infty} \gamma^t r(s_t, a_t) \right]. \tag{4}$$

In NGE, the mutated species continues the optimization by initializing the parameters with the parameters inherited from its parent species. In past work (Sims, 1994), species in one generation are trained separately for a fixed number of updates, which is biased and potentially undertrained or

overtrained. In next generations, new species have to discard old controllers if the graph topology is *different*, which might waste valuable computation resources.

## 3.2 POLICY REPRESENTATION

Given a species with graph $\mathcal{G}$, we train the parameters $\theta$ of policy network $\pi_\theta(a^t|s^t)$ using reinforcement learning. Similar to (Wang et al., 2018), we use a GNN as the policy network of the controller. A graphical representation of our model is shown in Figure 1. We follow notation in Section 2.2.

For the *input model*, we parse the input state vector $s^t$ obtained from the environment into a graph, where each node $u \in V$ fetches the corresponding observation $o(u, t)$ from $s^t$, and extracts the feature $x_u^{O,t}$ with an embedding function $\Phi$. We also encode the attribute information $A(u)$ into $x_u^A$ with an embedding function denoted as $\zeta$. The input feature $x_u^t$ is thus calculated as:

$$x_u^{O,t} = \Phi(o(u,t)), \;\; x_u^A = \zeta(A(u)),$$
$$x_u^t = [x_u^{O,t}; x_u^A], \tag{5}$$

where [.] denotes concatenation. We use $\theta_\Phi, \theta_\zeta$ to denote the weights of embedding functions.

The *propagation model* is described in Section 2.2. We recap the propagation model here briefly: Initial hidden state for node $u$ is denoted as $h_u^{t;0}$, which are initialized from hidden states from the last timestep $h_u^{t-1,\mathcal{T}}$ or simply zeros. $\mathcal{T}$ internal propagation steps are performed for each timestep, during each step (denoted as $\tau \leq \mathcal{T}$) of which, every node sends messages to its neighboring nodes, and aggregates the received messages. $h_u^{t;\tau+1}$ is calculated by an update function that takes in $h_u^{t;\tau}$, node input feature $x_u^t$ and aggregated message $m_u^{t;\tau}$. We use summation as the aggregation function and a GRU (Chung et al., 2014) as the update function.

For the *output model*, we define the collection of controller nodes as $\mathcal{F}$, and define Gaussian distributions on each node's controller as follows:

$$\forall u \in \mathcal{F}, \;\; \mu_u^t = F_\mu(h_u^{t;\mathcal{T}}), \tag{6}$$
$$\sigma_u^t = F_\sigma(h_u^{t;\mathcal{T}}), \tag{7}$$

where $\mu_u$ and $\sigma_u$ are the mean and the standard deviation of the action distribution. The weights of output function are denoted as $\theta_F$. By combining all the actions produced by each node controller, we have the policy distribution of the agent:

$$\pi(a^t|s^t) = \prod_{u \in \mathcal{F}} \pi_u(a_u^t|s^t) = \prod_{u \in \mathcal{F}} \frac{1}{\sqrt{2\pi(\sigma_u^t)^2}} \exp\left(\frac{(a_u^t - \mu_u^t)^2}{2(\sigma_u^t)^2}\right) \tag{8}$$

We optimize $\pi(a^t|s^t)$ with PPO, the details of which are provided in Appendix A.

## 3.3 GRAPH MUTATION WITH UNCERTAINTY

Between generations, the graphs evolve from parents to children. We allow the following basic operations as the mutation primitives on the parent's graph $\mathcal{G}$:

$\mathcal{M}_1$, **Add-Node**: In the $\mathcal{M}_1$ (Add-Node) operation, the growing of a new body part is done by sampling a node $v \in V$ from the parent, and append a new node $u$ to it. We randomly initialize $u$'s attributes from an uniform distribution in the attribute space.

$\mathcal{M}_2$, **Add-Graph**: The $\mathcal{M}_2$ (Add-Graph) operation allows for faster evolution by reusing the sub-trees in the graph with good functionality. We sample a sub-graph or leaf node $\mathcal{G}' = (V', E', A')$ from the current graph, and a placement node $u \in V(\mathcal{G})$ to which to append $\mathcal{G}'$. We randomly mirror the attributes of the root node in $\mathcal{G}'$ to incorporate a symmetry prior.

$\mathcal{M}_3$, **Del-Graph**: The process of removing body parts is defined as $\mathcal{M}_3$ (Del-Graph) operation. In this operation, a sub-graph $\mathcal{G}'$ from $\mathcal{G}$ is sampled and removed from $\mathcal{G}$.

$\mathcal{M}_4$, **Pert-Graph**: In the $\mathcal{M}_4$ (Pert-Graph) operation, we randomly sample a sub-graph $\mathcal{G}'$ and recursively perturb the parameter of each node $u \in V(\mathcal{G}')$ by adding Gaussian noise to $A(u)$.

We visualize a pair of example fish in Figure 1. The fish in the top-right is mutated from the fish in the top-left by applying $\mathcal{M}_1$. The new node (2) is colored magenta in the figure. To mutate each new

candidate graph, we sample the operation $\mathcal{M}$ and apply $\mathcal{M}$ on $\mathcal{G}$ as

$$\mathcal{G}' = \mathcal{M}(\mathcal{G}), \text{ where } \mathcal{M} \in \{\mathcal{M}_l, \, l = 1, 2, 3, 4\}, \, \mathrm{P}(\mathcal{M} = \mathcal{M}_l) = p_m^l. \tag{9}$$

$p_m^l$ is the probability of sampling each operation with $\sum_l p_m^l = 1$.

To facilitate evolution, we want to avoid wasting computation resources on species with low expected fitness, while encouraging NGE to test species with high uncertainty. We again employ a GNN to predict the fitness of the graph $\mathcal{G}$, denoted as $\xi_P(\mathcal{G})$. The weights of this GNN are denoted as $\psi$. In particular, we predict the AF score with a similar propagation model as our policy network, but the observation feature is only $x_u^A$, i.e., the embedding of the attributes. The output model is a graph-level output (as opposed to node-level used in our policy), regressing to the score $\xi$. After each generation, we train the regression model using the L2 loss.

However, pruning the species greedily may easily overfit the model to the existing species since there is no modeling of uncertainty. We thus propose Graph Mutation with Uncertainty (GM-UC) based on Thompson Sampling to balance between exploration and exploitation. We denote the dataset of past species and their AF score as $\mathcal{D}$. GM-UC selects the best graph candidates by considering the posterior distribution of the surrogate $P(\psi | \mathcal{D})$:

$$\mathcal{G}^* = \arg\max_{\mathcal{G}} \, \mathbb{E}_{P(\psi | \mathcal{D})} \left[ \xi_P(\mathcal{G} | \psi) \right]. \tag{10}$$

Instead of sampling the full model with $\widetilde{\psi} \sim P(\psi | \mathcal{D})$, we follow Gal & Ghahramani (2016) and perform dropout during inference, which can be viewed as an approximate sampling from the model posterior. At the end of each generation, we randomly mutate $\mathcal{C} \geq \mathcal{N}$ new species from surviving species. We then sample a single dropout mask for the surrogate model and only keep $\mathcal{N}$ species with highest $\xi_P$. The details of GM-UC are given in Appendix F.

### 3.4 Rapid Adaptation using Policy Sharing

To leverage the transferability of GNNs across different graphs, we propose Policy Sharing (PS) to reuse old weights from parent species. The weights of a species in NGE are as follows:

$$\theta_G = (\theta_\Phi, \theta_\zeta, \theta_M, \theta_U, \theta_F), \tag{11}$$

where $\theta_\Phi, \theta_\zeta, \theta_M, \theta_U, \theta_F$ are the weights for the models we defined earlier in Section 3.2 and 2.2. Since our policy network is based on GNNs, as we can see from Figure 1, model weights of different graphs share the same cardinality (shape). A different graph will only alter the paths of message propagation. With PS, new species are provided with a strong weight initialization, and the evolution will less likely be dominated by species that are more ancient in the genealogy tree.

Previous approaches including naive evolutionary structure search (ESS-Sims) (Sims, 1994) or random graph search (RGS) utilize human-engineered one-layer neural network or a fully connected network, which cannot reuse controllers once the graph structure is changed, as the parameter space for $\theta$ might be different. And even when the parameters happen to be of the same shape, transfer learning with unstructured policy controllers is still hardly successful (Rajeswaran et al., 2017). We denote the old species in generation $j$, and its mutated species with *different* topologies as $(\theta_B^j, \mathcal{G})$, $(\theta_B^{j+1}, \mathcal{G}')$ in baseline algorithm ESS-Sims and RGS, and $(\theta_G^j, \mathcal{G})$, $(\theta_G^{j+1}, \mathcal{G}')$ for NGE. We also denote the network initialization scheme for fully-connected networks as $\mathcal{B}$. We show the parameter reuse between generations in Table 1.

| Algorithm | Mutation | Parameter Space | Policy Initialization |
|---|---|---|---|
| ESS-Sims, RGS | $\mathcal{G} \to \mathcal{G}'$ | $\{\theta_B(\mathcal{G})\} \cap \{\theta_B(\mathcal{G}')\} = \emptyset$ | $\theta_B^{j+1} \overset{init}{\sim} \mathcal{B}(\mathcal{G}')$, $\theta_B^j$ not reused |
| NGE | $\mathcal{G} \to \mathcal{G}'$ | $\{\theta_G(\mathcal{G})\} = \{\theta_G(\mathcal{G}')\}$ | $\theta_G^{j+1} \overset{init}{=} \theta_G^j$ |

Table 1: Parameter reuse between species and its mutated children if the topologies are different.

Figure 2: The performance of the graph search for RGS, ES and NGE. The figures on are the example creatures obtained from each of the method. The graph structure next to the figure are the corresponding graph structure. We included the original species for reference.

## 4 EXPERIMENTS

In this section, we demonstrate the effectiveness of NGE on various evolution tasks. In particular, we evaluate both, the most challenging problem of searching for the optimal body structure from scratch in Section 4.1, and also show a simpler yet useful problem where we aim to optimize human-engineered species in Section 4.2 using NGE. We also provide an ablation study on GM-UC in Section 4.3, and an ablation study on computational cost or generation size in Section 4.4.

Our experiments are simulated with MuJoCo. We design the following environments to test the algorithms. **Fish Env**: In the `fish` environment, graph consists of ellipsoids. The reward is the swimming-speed along the $y$-direction. We denote the reference human-engineered graph (Tassa et al., 2018) as $\mathcal{G}_F$. **Walker Env**: We also define a 2D environment `walker` constructed by cylinders, where the goal is to move along $x$-direction as fast as possible. We denote the reference human-engineered walker as $\mathcal{G}_W$ and cheetah as $\mathcal{G}_C$ (Tassa et al., 2018). To validate the effectiveness of NGE, baselines including previous approaches are compared. We do a grid search on the hyper-parameters as summarized in Appendix E, and show the averaged curve of each method. The baselines are introduced as follows:

**ESS-Sims**: This method was proposed in (Sims, 1994), and applied in (Cheney et al., 2014; Taylor, 2017), which has been the most classical and successful algorithm in automatic robotic design. In the original paper, the author uses evolutionary strategy to train a human-engineered one layer neural network, and randomly perturbs the graph after each generation. With the recent progress of robotics and reinforcement learning, we replace the network with a 3-layer Multilayer perceptron and train it with PPO instead of evolutionary strategy.

**ESS-Sims-AF**: In the original ESS-Sims, amortized fitness is not used. Although amortized fitness could not be fully applied, it could be applied among species with the same topology. We name this variant as ESS-Sims-AF.

**ESS-GM-UC**: ESS-GM-UC is a variant of ESS-Sims-AF, which combines GM-UC. The goal is to explore how GM-UC affects the performance without the use of a structured model like GNN.

**ESS-BodyShare**: We also want to answer the question of whether GNN is indeed needed. We use both an unstructured models like MLP, as well as a structured model by removing the message propagation model.

**RGS**: In the Random Graph Search (RGS) baseline, a large amount of graphs are generated randomly. RGS focuses on exploiting given structures, and does not utilize evolution to generate new graphs.

### 4.1 EVOLUTION TOPOLOGY SEARCH

In this experiment, the task is to evolve the graph and the controller from scratch. For both `fish` and `walker`, species are initialized as random $(\mathcal{G}, \theta)$. Computation cost is often a concern among structure search problems. In our comparison results, for fairness, we allocate the same computation budget to all methods, which is approximately 12 hours on a EC2 `m4.16xlarge` cluster with 64 cores for one session. A grid search over the hyper-parameters is performed (details in Appendix E). The averaged curves from different runs are shown in Figure 2. In both `fish` and `walker` environments, NGE is the best model. We find RGS is not able to efficiently search the space of $\mathcal{G}$ even after evaluating $12,800$ different graphs. The performance of ESS-Sims grows faster for the earlier

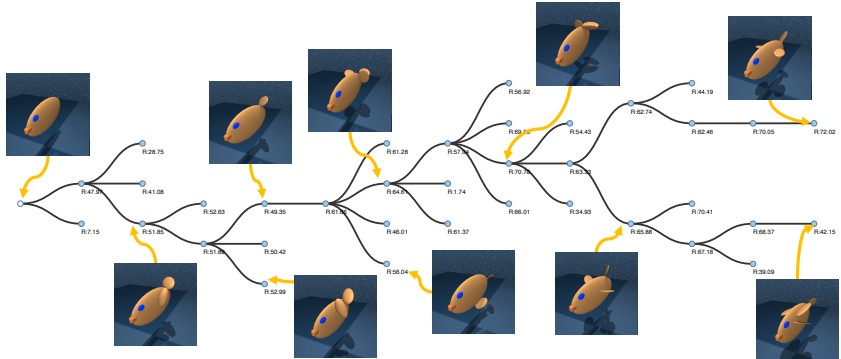

Figure 3: The genealogy tree generated using NGE for `fish`. The number next to the node is the reward (the averaged speed of the fish). For better visualization, we down-sample genealogy sub-chain of the winning species. NGE agents gradually grow symmetrical side-fins.

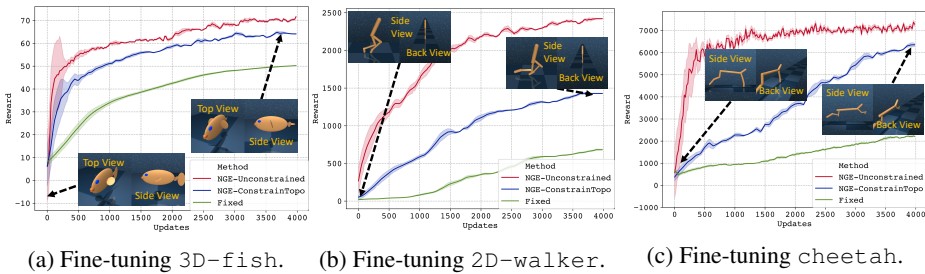

| (a) Fine-tuning `3D-fish`. | (b) Fine-tuning `2D-walker`. | (c) Fine-tuning `cheetah`. |

Figure 4: Fine-tuning results on different creatures compared with baseline where structure is fixed. The figures included the species looking from 2 different angles.

generations, but is significantly worse than our method in the end. The use of AF and GM-UC on ESS-Sims can improve the performance by a large margin, which indicates that the sub-modules in NGE are effective. By looking at the generated species, ESS-Sims and its variants overfit to local species that dominate the rest of generations. The results of ESS-BodyShare indicates that, the use of structured graph models without message passing might be insufficient in environments that require global features, for example, `walker`.

To better understand the evolution process, we visualize the genealogy tree of fish using our model in Figure 3. Our fish species gradually generates three fins with preferred $\{A(u)\}$, with two side-fins symmetrical about the fish torso, and one tail-fin lying in the middle line. We obtain similar results for `walker`, as shown in Appendix C. To the best of our knowledge, our algorithm is the first to automatically discover kinematically plausible robotic graph structures.

## 4.2 FINE-TUNING SPECIES

Evolving every species from scratch is costly in practice. For many locomotion control tasks, we already have a decent human-engineered robot as a starting point. In the fine-tuning task, we verify the ability of NGE to improve upon the human-engineered design. We showcase both, unconstrained experiments with NGE where the graph $(V, E, A)$ is fine-tuned, and constrained fine-tuning experiments where the topology of the graph is preserved and only the node attributes $\{A(u)\}$ are fine-tuned. In the baseline models, the graph $(V, E, A)$ is fixed, and only the controllers are trained. We can see in Figure 4 that when given the same wall-clock time, it is better to co-evolve the attributes and controllers with NGE than only training the controllers.

The figure shows that with NGE, the cheetah gradually transforms the forefoot into a *claw*, the 3D-fish rotates the pose of the side-fins and tail, and the 2D-walker evolves bigger feet. In general, unconstrained fine-tuning with NGE leads to better performance, but not necessarily preserves the initial structures.

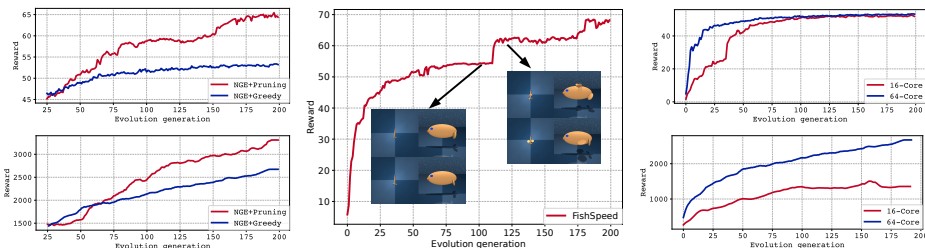

(a) Results of NGE with and without uncertainty. (b) Rapid graph evolution in the `fish` environment (c) The results of using different computation resource.

Figure 5: Results of ablation study, NGE without uncertainty results and rapid evolution during experiments.

### 4.3 GREEDY SEARCH V.S. EXPLORATION UNDER UNCERTAINTY

We also investigate the performance of NGE with and without Graph Mutation with Uncertainty, whose hyper-parameters are summarized in Appendix E. In Figure 5a, we applied GM-UC to the evolution graph search task. The final performance of the GM-UC outperforms the baseline on both `fish` and `walker` environments. The proposed GM-UC is able to better explore the graph space, showcasing its importance.

### 4.4 COMPUTATION COST AND GENERATION SIZE

We also investigate how the generation size $\mathcal{N}$ affect the final performance of NGE. We note that as we increase the generation size and the computing resources, NGE achieves marginal improvement on the simple `Fish` task. A NGE session with 16-core `m5.4xlarge` ($0.768 per Hr) AWS machine can achieve almost the same performance with 64-core `m4.16xlarge` ($3.20 per Hr) in `Fish` environment in the same wall-clock time. However, we do notice that there is a trade off between computational resources and performance for the more difficult task. In general, NGE is effective even when the computing resources are limited and it significantly outperforms RGS and ES by using only a small generation size of 16.

## 5 DISCUSSION

In this paper, we introduced NGE, an efficient graph search algorithm for automatic robot design that co-evolves the robot design graph and its controllers. NGE greatly reduces evaluation cost by transferring the learned GNN-based control policy from previous generations, and better explores the search space by incorporating model uncertainties. Our experiments show that the search over the robotic body structures is challenging, where both random graph search and evolutionary strategy fail to discover meaning robot designs. NGE significantly outperforms the naive approaches in both the final performance and computation time by an order of magnitude, and is the first algorithm that can discovers graphs similar to carefully hand-engineered design. We believe this work is an important step towards automated robot design, and may show itself useful to other graph search problems.

**Acknowledgements** Partially supported by Samsung and NSERC. We also thank NVIDIA for their donation of GPUs.

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

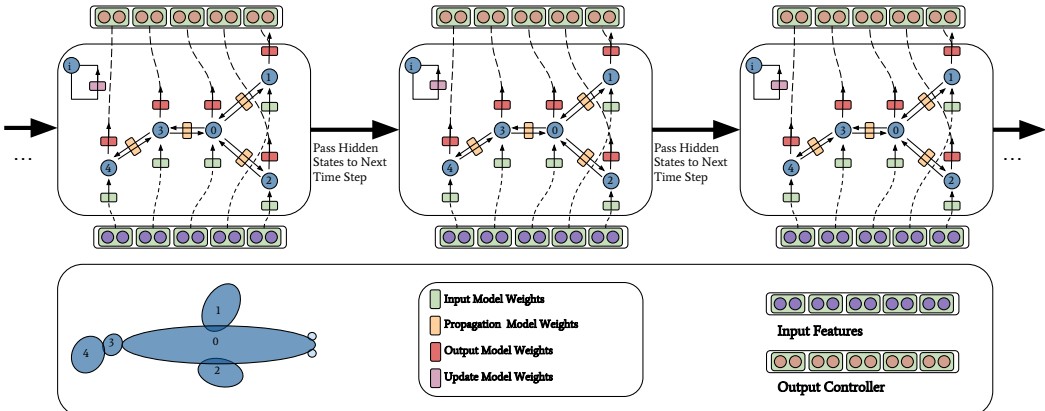

Figure 6: In this figure, we show the computation graph of NerveNet++. At each timestep, every node in the graph updates its hidden state by absorbing the messages as well as the input feature. The output function takes the hidden states as input and outputs the controller (or policy) of the agent.

## A  DETAILS OF NERVENET++

Similar to NerveNet, we parse the agent into a graph, where each node in the graph corresponds to the physical body part of the agents. For example, the `fish` in Figure 1 can be parsed into a graph of five nodes, namely the torso (0), left-fin (1), right-fin (2), and tail-fin bodies (3, 4). By replacing MLP with NerveNet, the learnt policy has much better performance in terms of robustness and the transfer learning ability. We here propose minor but effective modifications to Wang et al. (2018), and refer to this model as NerveNet++.

In the original NerveNet, at every timestep, several propagation steps need to be performed such that every node is able to receive global information before producing the control signal. This is time and memory consuming, with the minimum number of propagation steps constrained by the depth of the graph.

Since the episode of each game usually lasts for several hundred timesteps, it is computationally expensive and ineffective to build the full back-propagation graph. Inspired by Mnih et al. (2016), we employ the truncated graph back-propagation to optimize the policy. NerveNet++ is suitable for an evolutionary search or population-based optimization, as it brings speed-up in wall-clock time, and decreases the amount of memory usage.

Therefore in NerveNet++, we propose a propagation model with the memory state, where each node updates its hidden state by absorbing the input feature and a message with time. The number of propagation steps is no longer constrained by the depth of the graph, and in back-propagation, we save memory and time consumption with truncated computation graph.

The computational performance evaluation is provided in Appendix B. NerveNet++ model is trained by the PPO algorithm Schulman et al. (2017); Heess et al. (2017),

## B  OPTIMIZATION WITH TRUNCATED BACKPROPAGATION

During training, the agent generates the rollout data by sampling from the distribution $a_t \sim \pi(a_t|s_t)$ and stores the training data of $\mathcal{D} = \{a^t, s^t, \{h_u^{t;\tau=0}\}\}$. To train the reinforcement learning agents with memory, the original training objective is

$$J(\theta) = \mathbb{E}_\pi \left[ \sum_{t=0}^{\infty} \gamma^t r(s^t, a^t, \{h_u^{t,\tau=0}\}) \right], \tag{12}$$

where we denote the whole update model as $H$ and

$$h_u^{t+1,\tau=0} = H(\{h_v^{t,\tau=0}\}, s^t, a^t). \tag{13}$$

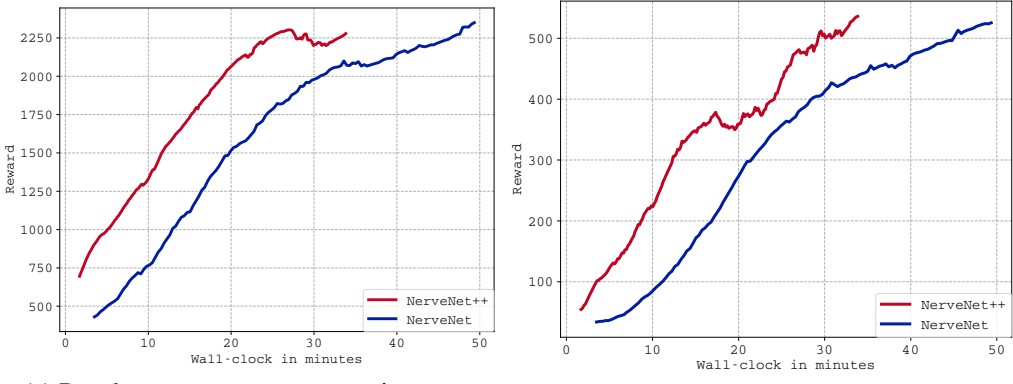

(a) Results on `Cheetah-V1` environment.

(b) Results on `Walker2d-V1` environment.

Figure 7: In these two figures, we show that to reach similar performance, NerveNet++ took shorter time comparing to original NerveNet.

The memory state $h_u^{t+1,\tau}$ depends on the previous actions, observations, and states. Therefore, the full back-propagation graph will be the same length as the episode length, which is very computationally intensive. The intuition from the authors in Mnih et al. (2016) is that, for the RL agents, the dependency of the agents on timesteps that are far-away from the current timestep is limited. Thus, negligible accuracy of the gradient estimator will be lost if we truncate the back-propagation graph. We define a back-propagation length $\Gamma$, and optimize the following objective function instead:

$$J_T(\theta) = \mathbb{E}_\pi \left[ \sum_{t=0}^{\infty} \sum_{\kappa=0}^{\Gamma-1} \gamma^{t+\kappa} r(s_{t+\kappa}, a_{t+\kappa}, \{h_u^{t,\tau=0}\}) \right], \text{ where} \tag{14}$$

$$h_u^{t+\kappa,\tau=0} = \begin{cases} H(\{h_v^{t+\kappa-1,\tau=0}, \forall v\}, s_{t+\kappa-1}, a_{t+\kappa-1}) & \kappa \neq 0, \\ h_u^{t,\tau=0} \in \mathcal{D} & \kappa = 0, \end{cases} \tag{15}$$

Essentially this optimization means that we only back-propagate up to $\Gamma$ timesteps, namely at the places where $\kappa = 0$, we treat the hidden state as input to the network and stop the gradient. To optimize the objective function, we follow same optimization procedure as in Wang et al. (2018), which is a variant of PPO Schulman et al. (2017), where a surrogate loss $J_{\text{ppo}}(\theta)$ is optimized. We refer the readers to these papers for algorithm details.

## C    FULL NGE RESULTS

Similar to the fish genealogy tree, in Fig. 8, the simple initial walking agent evolves into a cheetah-like structure, and is able to run with high speed. We also show the species generated by NGE, ESS-Sims (ESS-Sims-AF to be more specific, which has the best performance among all ESS-Sims variants.) and RGS.

## D    RESETTING CONTROLLER FOR FAIR COMPETITION

Although amortized fitness is a better estimation of the ground-truth fitness, it is still biased. Species that appear earlier in the experiment will be trained for more updates if it survives. Indeed, intuitively, it is possible that in real nature, species that appear earlier on will dominate the generation by number, and new species are eliminated even if the new species has better fitness. Therefore, we design the experiment where we reset the weights for all species $\theta = (\theta_\Phi, \theta_\zeta, \theta_M, \theta_U, \theta_F)$ randomly. By doing this, we are forcing the species to compete fairly. From Fig 10, we notice that this method helps exploration, which leads to a higher reward in the end. However, it usually takes a longer time for the algorithm to converge. Therefore for the graph search task in Fig 2, we do not include the results with the controller-resetting.

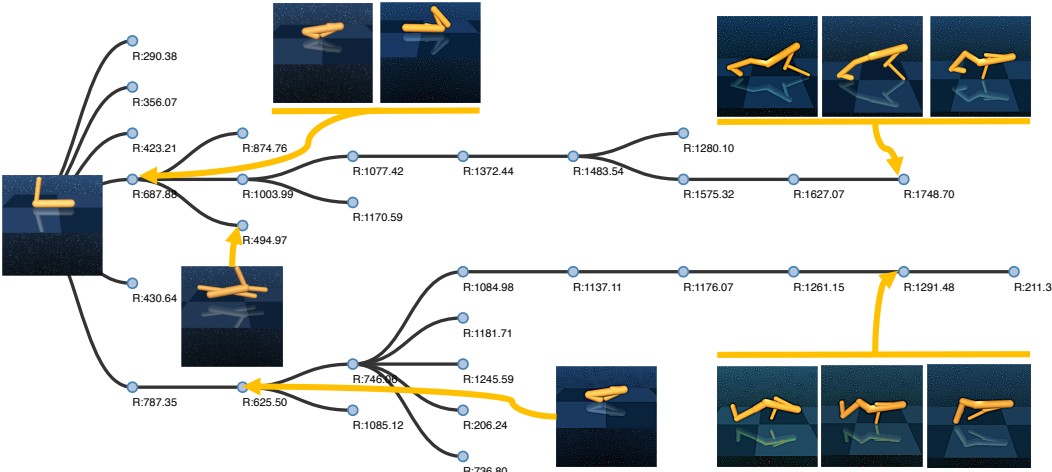

Figure 8: Our `walker` species gradually grows two foot-like structures from randomly initialized body graph.

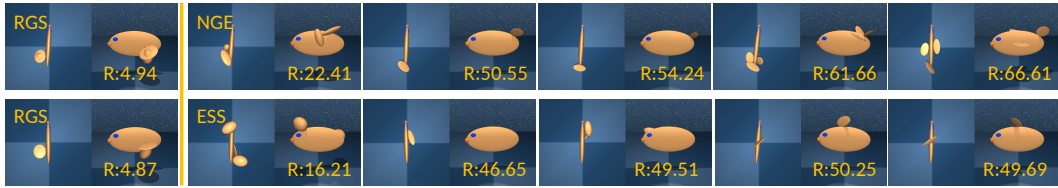

Figure 9: We present qualitative comparison between the three algorithms in the figure. Specifically, the aligned comparison between our method and naive baseline are the representative creatures at the same generation (using same computation resources). Our algorithm notably display stronger dominance in terms of its structure as well as reward.

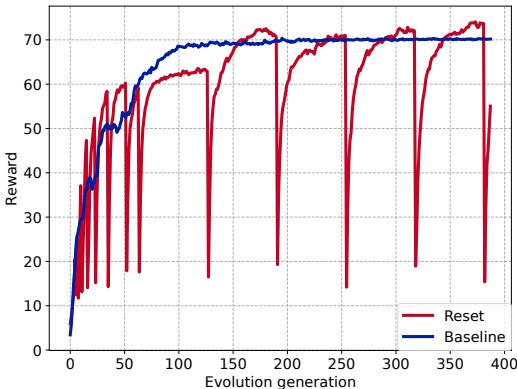

Figure 10: The results of resetting controller scheme and baselines.

## E  HYPER-PARAMETERS SEARCHED

All methods are given equal amount of computation budget. To be more specific, the number of total timesteps generated by all species for all generations is the same for all methods. For example, if we use 10 training epochs in one generation, each of the epoch with 2000 sampled timesteps, then the computation budget allows NGE to evolve for 200 generations, where each generation has a species size of 64. For NGE, RGS, ESS-Sims-AF models in Fig 11, we run a grid search over the hyper-parameters recorded in Table 2, and Table 3, and plot the curve with the best results respectively.

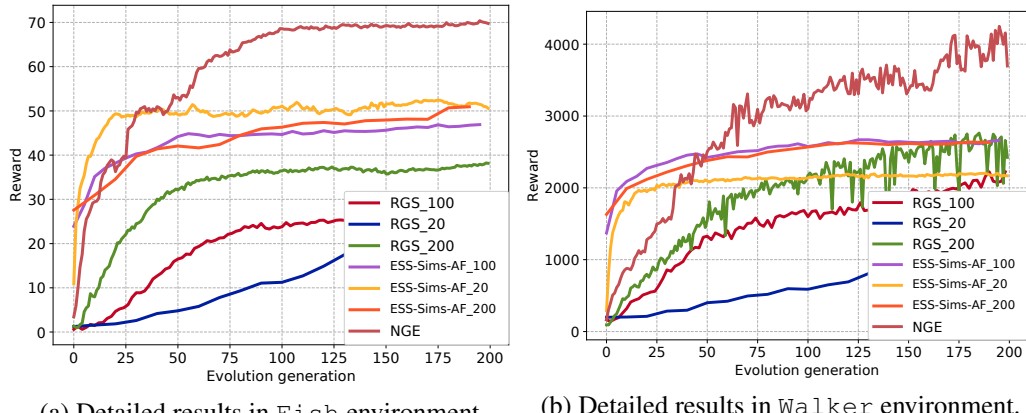

(a) Detailed results in `Fish` environment.

(b) Detailed results in `Walker` environment.

Figure 11: The results of the graph search

Since the number of generations for the RGS baseline can be regarded as 1, its curve is plotted with the number of updates normalized by the computation resource as x-axis.

Here we show the detail figures of six baselines, which are: RGS-20, RGS-100, RGS-200, and ESS-Sims-AF-20, ESS-Sims-AF-100, ESS-Sims-AF-200. The number attached to the baseline names indicates the number of inner-loop policy training epochs. In the case of RGS-20, where more than 12800 different graphs are searched over, the average reward is still very low. Increasing the number of inner-loop training of species to 100 and 200 does not help the final performance significantly.

To test the performance with and without GM-UC, we use 64-core clusters (generations of size 64). Here, the hyper-parameters are chosen to be the first value available in Table 2 and Table 3.

| Items | Value Tried |
|---|---|
| Number of Iteration Per Update | 10, 20, 100, 200 |
| Number of Species per Generation | 16, 32, 64, 100 |
| Elimination Rate | 0.15, 0.20, 0.3 |
| Discrete Socket | Yes, True |
| Timesteps per Updates | 2000, 4000, 6000 |
| Target KL | 0.01 |
| Learning Rate Schedule | Adaptive |
| Number of Maximum Generation | 400 |
| Prob of `Add-Node`, `Add-Graph` | 0.15 |
| Prob of `Pert-Graph` | 0.15 |
| Prob of `Del-Graph` | 0.15 |
| Allow Mirrowing Attrs in `Add-Graph` | Yes, No |
| Allow Resetting Controller | Yes, No |
| Resetting Controller Freq | 50, 100 |

Table 2: Hyperparameter grid search options.

## F   MODEL BASED SEARCH USING THOMPSON SAMPLING

Thompson Sampling is a simple heuristic search strategy that is typically applied to the multi-armed bandit problem. The main idea is to select an action proportional to the probability of the action being optimal. When applied to the graph search problem, Thompson Sampling allows the search to balance the trade-off between exploration and exploitation by maximizing the expected fitness under the posterior distribution of the surrogate model.

| Items | Value Tried |
|---|---|
| Allow `Graph-Add` | True, False |
| Graph Mutation with Uncertainty | True, False |
| Pruning Temperature | 0.01, 0.1, 1 |
| Network Structure | NerveNet, NerveNet++ |
| Number Candidates before Pruning | 200, 400 |

Table 3: Hyperparameters grid search options for NGE.

Formally, Thompson Sampling selects the best graph candidates at each round according to the expected estimated fitness $\xi_P$ using a surrogate model. The expectation is taken under the posterior distribution of the surrogate $P(\text{model}|\text{data})$:

$$\mathcal{G}^* = \arg\max_{\mathcal{G}} \mathbb{E}_{P(\text{model}|\text{data})} \left[ \xi_P(\mathcal{G}|\text{model}) \right]. \tag{16}$$

### F.1 SURROGATE MODEL ON GRAPHS.

Here we consider a graph neural network (GNN) surrogate model to predict the average fitness of a graph as a Gaussian distribution, namely $P(f(\mathcal{G})) \sim \mathcal{N}\left(\xi_P(\mathcal{G}), \sigma^2(\mathcal{G})\right)$. We use a simple architecture that predicts the mean of the Gaussian from the last hidden layer activations, $h_W(\mathcal{G}) \in \mathbb{R}^D$, of the GNN, where $W$ are the weights in the GNN up to the last hidden layer.

**Greedy search.** We denoted the size of dataset as $N$. The GNN weights are trained to predict the average fitness of the graph as a standard regression task:

$$\min_{W, W_{out}} \frac{\beta}{2} \sum_{n=1}^{N} \left( \xi(\mathcal{G}_n) - \xi_P(\mathcal{G}_n) \right)^2, \quad \text{where} \quad \xi_P(\mathcal{G}_n) = W_{out}^T h_W(\mathcal{G}_n) \tag{17}$$

---

**Algorithm 2** Greedy Search

---

1: Initialize generation $\mathcal{P}^0$
2: **for** $j <$ maximum generations **do**
3:      Collect the $(\xi_i^k, \mathcal{G}_i^k)$ from previous $k \leq j$ generations          ▷ Update dataset
4:      Train $W$ and $W_{out}$ on $\{(\xi_i^k, \mathcal{G}_i^k)\}_{n=1}^N$          ▷ Train GM-UC
5:      Propose $\mathcal{C}$ new graph $\{\mathcal{G}_i\}_{i=1}^{\mathcal{C}}, \mathcal{C} >> M$.          ▷ Propose new candidates
6:      Rank $\{\xi_P(\mathcal{G}_i|W, W_{out})\}_{i=1}^{\mathcal{C}}$ on the proposals and pick the top $\mathcal{K}$      ▷ Prune candidates
7:      Update generation $\mathcal{P}^j$
8:      **for** $m < \mathcal{N}$ **do**          ▷ Train and evaluate each species
9:          **for** $k <$ maximum parameter updates **do**
10:              Train policy $\pi_{\mathcal{G}_m}$
11:          **end for**
12:          Evaluate the fitness $\xi(\mathcal{G}_m, \theta_m)$
13:      **end for**
14: **end for**

---

**Thompson Sampling** In practice, Thompson Sampling is very similar to the previous greedy search algorithm. Instead of picking the top action according to the best model parameters, at each generation, it draws a sample of the model and takes a greedy action under the sampled model.

**Approximating Thompson Sampling using Dropout** Performing dropout during inference can be viewed as an approximately sampling from the model posterior. At each generation, we will sample a single dropout mask for the surrogate model and rank all the proposed graphs accordingly.

---

**Algorithm 3** Thompson Sampling using Bayesian Neural Networks

---

1: Initialize generation $\mathcal{P}^0$
2: **for** $j <$ maximum generations **do**
3:     Collect the $(\xi_i^k, \mathcal{G}_i^k)$ from previous $k \leq j$ generations         ▷ Update dataset
4:     Train $W$ and $W_{out}$ on $\{(\xi_i^k, \mathcal{G}_i^k)\}_{n=1}^N$         ▷ Train GM-UC
5:     Propose $\mathcal{C}$ new graph $\{\mathcal{G}_i\}_{i=1}^{\mathcal{C}}, \mathcal{C} >> M$.         ▷ Propose new candidates
6:     Sample a model from the posterior of the weights.
7:         e.g. $\widetilde{W}, \widetilde{W}_{out} \sim P(W, W_{out}|\mathcal{D}) \approx \mathcal{N}([W, W_{out}], [W, W_{out}])$
8:         (similar to DropConnect Wan et al. (2013))
9:     Rank $\{\xi_P(\mathcal{G}_i|\widetilde{W}, \widetilde{W}_{out})\}_{i=1}^{\mathcal{C}}$ on the proposals and pick the top $\mathcal{K}$
10:     **for** $m < \mathcal{N}$ **do**         ▷ Train and evaluate each species
11:         **for** $k <$ maximum parameter updates **do**
12:             Train policy $\pi_{\mathcal{G}_m}$
13:         **end for**
14:         Evaluate the fitness $\xi(\mathcal{G}_m, \theta_m)$
15:     **end for**
16: **end for**

---

---

**Algorithm 4** Thompson Sampling with Dropout

---

1: Initialize generation $\mathcal{P}^0$
2: **for** $j <$ maximum generations **do**
3:     Collect the $(\xi_i^k, \mathcal{G}_i^k)$ from previous $k \leq j$ generations         ▷ Update dataset
4:     Train $W$ and $W_{out}$ on $\{\mathcal{G}_n, \xi(\mathcal{G}_n)\}_{n=1}^N$ using dropout rate 0.5 on the inputs of the fc layers.
5:     Propose $\mathcal{C}$ new graph $\{\mathcal{G}_i\}_{i=1}^{\mathcal{C}}, \mathcal{C} >> M$.         ▷ Propose new candidates
6:     Sample a dropout mask $m_i$ for the hidden units
7:     Rank $\{\xi_P(\mathcal{G}_i|W, W_{out}, m_i)\}_{i=1}^J$ on the proposals and pick the top $\mathcal{K}$
8:     **for** $m < \mathcal{N}$ **do**         ▷ Train and evaluate each species
9:         **for** $k <$ maximum parameter updates **do**
10:             Train policy $\pi_{\mathcal{G}_m}$
11:         **end for**
12:         Evaluate the fitness $\xi(\mathcal{G}_m, \theta_m)$
13:     **end for**
14: **end for**

---

