# OpenReview forum: "Neural Graph Evolution: Towards Efficient Automatic Robot Design"
_ICLR.cc/2019/Conference_

### Official Review · AnonReviewer1 · 2018-11-04
**Direct application of ES with NerveNet for fitness evaluation**

**Rating:** 6
**Confidence:** 4

**Review:**

[Summary]:
This paper tackles the problem of automatic robot design. The most popular approach to doing this has been evolutionary methods which work by evolving morphology of agents in a feed-forward manner using a propagation and mutation rules. This is a non-differentiable process and relies on maintaining a large pool of candidates out of which best ones are chosen with the highest fitness. In robot design for a given task using rewards, training each robot design using RL with rewards is an expensive process and not scalable. This paper uses graph network to train each morphology using RL. Thereby, allowing the controller to share parameters and reuse information across generations. This expedites the score function evaluation improving the time complexity of the evolutionary process.

[Strengths]:
This paper shows some promise when graph network-based controllers augmented with evolutionary algorithms. Paper is quite easy to follow.

[Weaknesses and Clarifications]:
=> Robot design area has been explored extensively in classical work of Sims (1994) etc. using ES. Given that, the novelty of the paper is fairly incremental as it uses NerveNet to evaluate fitness and ES for the main design search.
=> Environment: The experimental section of the paper can be further improved. The approach is evaluated only in three cases: fish, walker, cheetah. Can it be applied to more complex morphologies? Humanoid etc. maybe?
=> Baselines: The comparison provided in the paper is weak. At first, it compares to random graph search and ES. But there are better baselines possible. One such example would be to have a network for each body part and share parameters across each body part. This network takes some identifying information (ID, shape etc.) about body part as input. As more body parts are added, more such network modules can be added. How would the given graph network compare to this? This baseline can be thought of a shared parameter graph with no message passing.
=> The results shown in Figure-4 (Section-4.2) seems unclear to me. As far as I understand, the model starts with hand-engineered design and then finetuned using evolutionary process. However, the original performance of the hand-engineered design is surprisingly bad (see first data point in any plot in Figure-4). Does the controller also start from scratch? If so, why? Also, it is not clear what is the meaning of generations if the graph is fixed, can't it be learned altogether at once?

[Recommendation]:
I request the authors to address the comments raised above. Overall, this is a reasonable paper but experimental section needs much more attention.

---

> ### Author Response · Authors · 2018-11-19
> **Response to Reviewer 1**
>
> We thank the reviewer for the suggestions.
>
> Q1: Robot design were explored in (Sims, 1994) etc. The novelty of the paper is fairly incremental.
>
> We respectfully disagree and believe our contributions are significant. We note that only NGE among all the baselines has the ability to optimize both the graph G and the controller parameters. Graph neural network formulation is KEY here, enabling it to perform this efficient policy transfer. To the best of our knowledge, the traditional methods (such as (Sims, 1994)) require re-optimizing parameters of the controllers from scratch for each different topologies, which is computationally demanding and breaks the joint-optimization.
>
> To further showcase our work with respect to prior art, we added (Sims, 1994) as an additional baseline in the latest revision. We refer the reviewer to the general response for details. NGE has about 2x performance of (Sims, 1994) in both fish and walker environments. Moreover, we argue the videos of (Sims, 1994) might be confusing as it mixes the results of policy evolution from human-designed robots and structure evolution.
>
> Q2: Can it be applied to more complex morphologies? Humanoid etc. maybe?
> NGE can be applied to evolve humanoids, however, there are two major difficulties in doing that in practice.
> 1. Training humanoid controllers is of orders of magnitude more difficult than training cheetah (Schulman, 2017).
> 2. To evolve realistic humanoid structure (e.g. hands, symmetrical limbs), one would need to have more realistic environments that better reflect tasks and complexity in the real world.
> However, we agree that this is a very interesting direction for the future.
>
> Q3: Comparison to more baseline, for example models with no message passing.
>
> We thank the reviewer for pointing out the baseline of no message passing in GNN, which we named as ESS-BodyShare.
>
> In the latest revision, we have 5 baselines from previous research and modern variants, which further showcases the significance of our work. In general, NGE has significant improvement both quantitatively and qualitatively. We refer the reviewer to the general response for further information.
>
> Specifically for ESS-BodyShare baseline:
> 	       |  NGE     | ESS-BodyShare
> fish         |  70.21   |  54.97    (78.3% of NGE)
> Walker   |  4157.9 |   2185.1 (52.5% of NGE)
>
> In environment where global information is needed (for example, walker with multiple rigid body contact), the performance is jeopardized. But in an easier environment, message passing is less needed.
>
>
> Q4: Clarification of Figure-4 (Section-4.2)
>
> Our aim was to show that in the case where the human-engineered topology needs to be preserved, it is better to co-evolve the attributes and controllers with NGE rather than only training the controllers (controllers are trained from scratch for both NGE and baselines).
>
> The x-axis was scaled according to the number of updates. We apologize for the lack of clarity. We revised the x-axis from “generations” to parameter “updates” in the latest revision.
>
> In the latest revision, we also included the curve where the topologies are allowed to be changed, which leads to better performance, but does not necessarily preserve the initial structure.
>
> Schulman, 2017. "Proximal policy optimization algorithms." arXiv preprint arXiv:1707.06347 (2017).

---

> > ### Comment · AnonReviewer1 · 2018-12-08
> > **Response to rebuttal**
> >
> > Thank you for updating the paper with correct axis labels. Overall, I still feel the experiment section is very weak and the results are only shown in a few selected environments. Hence, I keep my review to be same, i.e., 6.

---

> > > ### Author Response · Authors · 2018-12-10
> > > **We thank the reviewer for the responses**
> > >
> > > We respect the reviewer's opinion and thanks again for the response. But still, we disagree with the claim that the experiment part is weak.
> > >
> > > In terms of the quality of baselines, we already include 5 comparing baselines including previous state-of-the-art. And NGE has the best performance and efficiency by a large margin (2x of previous state-of-the-art).
> > >
> > > The problem is novel / under-explored and there is no existing benchmark.
> > > We put in significant efforts to design 2 structure search environments and 3 fine-tuning environments, which requires weeks (even months) of engineering (robotics xml parser, graph xml generator, forward-kinematics, states mapping, etc.). We will release the code and environments after the reviewing period.
> > >
> > > We argue the evaluation of research should not be constrained by the number of experiments. And more focus can be paid on the novelty of algorithms and the inspiration that can be brought to the community.
> > >
> > > We would like to emphasize that our experiments show that, in the high-fidelity simulation like MuJoCo (previous research is conducted either in 2D environments or with simplified self-made engine), no previous approach can efficiently search for athletic walker or swimmer structures.
> > > Unlike the previous approaches that optimize the graph and the controllers separately, our proposed method jointly optimize discrete graph structure and the continuous controller parameters at the same time. Our joint optimization is a novel formulation, and effective approach that outperforms all the other baseline methods.
> > >
> > > This paper lies in the intersection of graph learning, reinforcement learning, robotics and structure search. Although it is a small step towards automatic robot structure search, we believe it will inspire following work in robotics, graph generation and neural architecture search.

---

### Official Review · AnonReviewer3 · 2018-11-08
**Interesting paper on co-optimizing robot structure and control**

**Rating:** 8
**Confidence:** 4

**Review:**

This paper discusses the optimization of robot structures, combined with their controllers. The authors propose a scheme
based on a graph representation of the robot structure, and a graph-neural-network as controllers. The experiments show
that the proposed scheme is able to produce walking and swimming robots in simulation. The results in this paper are impressive, and the paper seems free of technical errors.

The main criticism I have is that I found the paper harder to read. In particular, the exact difference between the proposed method and the ES baseline is not as clear as it could be. This makes the contribution of this paper in terms of the method
hard to judge. Please include further description of the ES cost function and algorithm in the main body of the paper.

The second point is that the proposed approach seems to modify a few things from the ES baseline. The efficacy of the separate modifications should be tested. Therefore I would like to see experiments with the ES cost function, but with
inclusion of the pruning step, and experiments with the AF-function but without the pruning step.

---

> ### Author Response · Authors · 2018-11-19
> **Response to Reviewer 3**
>
> We thank the reviewer for the reading and suggestions of our paper.
>
> Q1: The exact difference between the proposed method and the ES baseline is not as clear as it could be.
>
> We agree and apologize for the lack of clarity in some parts of our paper. We renamed all the models based on the original papers and their properties. We refer the reviewer to general response for further details of each baseline algorithms.
> We also improved clarity in the revised version.
>
> Q2: The second point is that the proposed approach seems to modify a few things from the ES baseline.
>
> We thank the reviewer for the insightful suggestion. In the latest version, to test the efficacy of each submodule of NGE, the baselines now include the algorithm with the inclusion of the pruning step, and the algorithms with AF and without AF using MLP.
>
> More specifically, the baselines are named:
>
> 1. ESS-Sims
> It is the baseline algorithm without the use of AF, as use by (Sims, 1994), (Cheney, 2014) and (Taylor, 2017).
> 2. ESS-Sims-AF
> The modern variant of ESS-Sims with the inclusion of AF.
> 3. ESS-GM-UC
> The modern variant of ESS-Sims with the inclusion of AF and graph mutation with uncertainty (pruning).
> For this baseline, we included the pruning module on top of ESS-Sims-AF. Similar to the original baselines available, we performed a grid search of hyperparameters and plot the average performance of the best set of hyperparameters.
>
> 	       |        NGE         | ESS-Sims  |  ESS-Sims-AF  | ESS-GM-UC | ESS-BodyShare |  RGS
> fish        |  **70.21**    |    38.32      |       51.24         |      54.40        |          54.97         |  20.96
> Walker  |  **4157.9**  |    1804.4    |      2486.9        |     2458.19     |          2185.1       |  1777.3
>
> Notice that GM-UC has a lower performance gain with the fully-connected network (ESS-Sims) than with GNN. We speculate that this happens in ESS-Sims because the controller is less dependent on the graph structure, and thus the fitness does not well capture the information about the topology. Thus, GM-UC is not able to extract as much information as with GNN.
>
> On the other hand, the use of AF can greatly affect the performance. The previous approach ESS-Sims can only get 38.32 / 1804 average final reward for fish and walker, respectively. The performance of walker is even very close to random graph search with no evolution. With the help of AF, the performance increases from 38.32 to 51.24 and 1804.4 to 2486.9, respectively.

---

> > ### Comment · AnonReviewer3 · 2018-12-10
> > **Response to rebuttal**
> >
> > The response makes the paper clearer. The added comparisons are interesting, although they could be more in depth. I keep my response as it was, due to the interesting proposed approach, and the obtained results.

---

### Official Review · AnonReviewer2 · 2018-11-09
**Interesting approach, inconclusive experiments**

**Rating:** 5
**Confidence:** 4

**Review:**

This paper proposes an approach for automatic robot design based on Neural graph evolution.
The overall approach has a flavor of genetical algorithms, as it also performs evolutionary operations on the graph, but it also allows for a better mechanism for policy sharing across the different topologies, which is nice.

My main concern about the paper is that, currently, the experiments do not include any strong baseline (the ES currently is not a strong baseline, see comments below).
The experiments currently demonstrate that optimizing both controller and hardware is better than optimizing just the controller, which is not surprising and is a phenomenon which has been previously studied in the literature.
What instead is missing is an answer to the question: Is it worth using a neural graph? what are the advantages and disadvantages compared to previous approaches?
I would like to see additional experiments to answer this questions.

In particular, I believe that any algorithms you compare against, you should optimize both G and theta, since optimizing purely the hardware is unfair.
You should use an existing ES implementation (e.g., from some well-known package) instead of a naive implementation, and as additional baseline also CMA-ES.
If you can also compare against one or two algorithms of your choice from the recent literature it would also give more value to the comparison.

Detailed comments:
- in the abstract you say that "NGE is the first algorithm that can automatically discover complex robotic graph structures". This statement is ambiguous and potentially unsupported by evidence. how do you define complex? that can or that did discover?
- in the introduction you mention that automatic robot design had limited success. This is rather subject, and I would tend to disagree.  Moreover, the same limitations that apply to other algorithms to make them successful, in my opinion, apply to your proposed algorithm (e.g., difficulty to move from simulated to real-world).
- The digression at the bottom of the first page about neural architecture search seem out of context and interrupts the flow of the introduction. What is the point that you are trying to make? Also, note that some of the algorithms that you are citing there have indeed applied beyond architecture search, eg. Bayesian optimization is used for gait optimization in robotics, and Genetic algorithms have been used for automatic robot design.
- The stated contributions number 3 and 5 are not truly contributions. #3 is so generic that a large part of the previous literature on the topic fall under this category -- not new. #5 is weak, and tell us more about the limitations of random search and naive ES than necessarily a merit of your approach.
- Sec 2.2: "(GNNs) are very effective" effective at what? what is the metric that you consider?
- Sec 3 "(PS), where weights are reused" can you already go into more details or refer to later sections?
- First line page 4 you mention AF, without introducing the acronym ever before.
- Sec 3.1: the statements about MB and MF algorithms are inaccurate. Model-based RL algorithms can work in real-time (e.g. http://proceedings.mlr.press/v78/drews17a/drews17a.pdf) and have been shown to have same asymptotic performance of MB controllers for simple robot control (e.g. https://arxiv.org/abs/1805.12114)
- "to speed up and trade off between evaluating fitness and evolving new species" Unclear sentence. speed up what? why is this a trade-off?
- Sec 3.4 can you recap all the parameters after eq.11? going through Sec 3.2 and 2.2 to find them is quite annoying.
- Sec 4.1:  would argue that computational cost is rarely a concern among evolutionary algorithms. The cost of evaluating the function is typically more pressing, and as a result it is important to have algorithms that can converge within a small number of iterations/generations.
- Providing the same computational budget seem rather arbitrary at the moment, and it heavily depends from implementation. How many evaluations do you perform for each method? why not having the same budget of experiments?

---

> ### Author Response · Authors · 2018-11-19
> **Response to Reviewer 2**
>
> We are afraid that there seems to be some confusion regarding our paper.  We apologize if this is caused by the lack of clarity in the use of abbreviation “ES” (see general response). In the latest revision, “Evolutionary structure search” is abbreviated as “ESS” for clarity. We emphasize that in the paper, NO “evolutionary strategy” but rather PPO is used to train the policy (see Section 2.1 and 3.2).
>
> We hope the reviewer can take time to revisit the paper in the light of this inconsistency. Also, we now have 5 baselines from previous research and modern variants, which we believe further showcases our contributions.
>
> Q1: The experiments do not include any strong baseline
>
> We added more baselines to further strengthen the significance of our work with respect to the previous approaches.
>
> The baselines now include (a)“ESS-Sims” (Sims, 1994), (Cheney, 2014), (Taylor, 2017), (b) ESS-Sims-AF, (c) ESS-GM-UC, (d) ESS-BodyShare and (5) Random graph search. We refer to the details of each baseline in the general response.
>
> 	        |      NGE         | ESS-Sims  | ESS-Sims-AF  | ESS-GM-UC | ESS-BodyShare |  RGS
> fish         | **70.21**    |    38.32     |       51.24         |      54.40       |          54.97         |  20.96
> Walker   |  **4157.9** |    1804.4   |      2486.9        |     2458.19   |          2185.1        |  1777.3
>
> The results show that NGE is significantly better than previous approaches and baselines. We did an ablation study by sequentially adding each sub-module of NGE separately. The table shows that submodules are effective and increase the performance of graph search.
>
> Q2: a) Optimizing both the controller and the hardware has been previously studied in the literature. Is it worth using a neural graph? b) All algorithms should optimize both G and theta for a fair comparison.
>
> By “optimizing both G and theta”, we meant to indicate that the learned controllers can be transferred to the next generation even if the topologies are changed (instead of throwing away old controllers). We note that only NGE among all the baselines has the ability to do that. Graph neural network formulation is KEY here, enabling it to perform this efficient policy transfer.
> To the best of our knowledge, the traditional methods require re-optimizing theta from scratch for each different topology, which is computationally demanding and breaks the joint-optimization.
> NGE approximately doubles the performance of previous approach (Sims, 1994) as shown in Q1.
>
> Please refer to Section 3.1 and Section 3.4 for more details.
>
> Q3: You should use an existing ES implementation (e.g., from some well-known package) instead of a naive implementation, and as additional baseline also CMA-ES.
>
> Again, we apologize for the confusing use of “ES” abbreviation. Evolutionary strategy is not used in the paper. We invite the reviewer to re-read our paper, since it seems to have led to a major misunderstanding.
> CMA-ES updates and utilize the covariance matrix of sampling distribution, which is not directly applicable to discrete structure optimization. We believe it will be a valuable future research direction.
>
> Q4: Providing the same computational budget seem rather arbitrary and depends on implementation.
>
> We are unsure what the reviewer is indicating, and would appreciate the additional clarification.
> In terms of the computational budget for each experiment, we compared different algorithms under different computational budget metrics, more specifically,  “wall-clock time”, “number of updates”, and the “final converged performance”. NGE performs best among all algorithms.
> We emphasize the fact that wall-clock time is a more common and realistic metric for comparing the structure search in practice.
>
> We agree that computational budget depends on implementation, and the curves in the paper are plotted based on the number of iterations/parameter update, which is independent of the implementation.
>
> Q5: The writing of the paper
>
> We sincerely thank the reviewer for the suggestions. We updated the changes in the latest version accordingly.

---

### Author Response · Authors · 2018-11-19
**General Response to the Reviewers**

We thank the reviewers for their response and suggestions. We have updated the paper and summarized the modifications here based on their feedback.

1. The abbreviation for “evolutionary structure search” is now changed from “ES” to “ESS” to reduce ambiguity. “ES” is abbreviated for “evolutionary search” and “evolutionary structure search” simultaneously in our original submission.

2. We rename “Graph Mutation (GM)” into “Graph Mutation with Uncertainty (GM-UC)”.

3. We added additional baselines from previous literature to benchmark the performance of our algorithm, and show that our proposed algorithm has significant improvement both quantitatively and qualitatively.

In particular, we added the following baselines:

a. ESS-Sims (Sims, 1994)
This method was proposed in (Sims, 1994), and applied in (Cheney, 2014), (Taylor, 2017), which has been the most classical and successful algorithm in automatic robotic design.
In the original paper, the author used evolutionary strategy to train a human-engineered one-layer neural network. With the recent progress of the robotics and reinforcement learning, we replaced the network with a 3-layer MLP and trained it with PPO instead of evolutionary strategy.

b. ESS-Sims-AF
In the original (Sims, 1994), amortized fitness is not used.
Although amortized fitness could not be applied in ESS since the shape of network parameters is changing, amortized fitness could be applied among agents with the same topology. We named this variant of ESS-Sims as “ESS-Sims-AF”.
This algorithm is essentially the old “ES” baseline in the earliest revision of the paper.

c. ESS-GM-UC
“ESS-GM-UC” is a variant of “ESS-Sims-AF” combined with Graph Mutation with Uncertainty. We would also want to explore how will GM-UC affect the performance without the use of structured model like GNN.

d. ESS-BodyShare
We would also want to answer the question of whether the graph neural network is needed.
As suggested by Reviewer 3, besides unstructured models like fully-connected network, we designed a structured model by removing the message propagation mode and named it “ESS-BodyShare”

e. RGS (random graph search)
The same baseline as described in the earlier revision.

The final performance the NGE and baselines are now shown in Figure 2 in the latest revision, which we summarize as the following table.

	       |       NGE           | ESS-Sims | ESS-Sims-AF  | ESS-GM-UC   | ESS-BodyShare |  RGS
Fish        |  ** 70.21  ** |    38.32     |       51.24         |      54.40         |          54.97          |  20.96
Walker   | ** 4157.9 ** |    1804.4   |      2486.9        |     2458.19     |          2185.1        |  1777.3

The results show that NGE is significantly better than previous approaches and baselines.

4. We improved the writing of the paper.
In particular, we added more literature review on related work as requested by the reviewers.
And we re-organized the writing of section 3.1, 3.2, 3.4, so that it is easier to understand and cause less confusion.

Sims, 1994, "Evolving virtual creatures." Proceedings of the 21st annual conference on Computer graphics and interactive techniques. ACM, 1994.

Cheney, 2014, et al. "Unshackling evolution: Evolving soft robots with multiple materials and a powerful generative encoding." ACM SIGEVOlution 7.1 (2014): 11-23.

Taylor, 2017. "Evolution in virtual worlds." arXiv preprint arXiv:1710.06055 (2017).

---

### Comment · Area_Chair1 · 2018-11-20
**Revised version available -- any updated opinions?**

Thanks to all for the detailed reviews and corresponding responses.

A revised version has been posted. There is also a useful "Compare Revisions" choice when you get to the Revisions page.

It would be good to hear from the reviewers if their concerns have been addressed, and if they are going to make any score revisions.  There is still some disparity, mainly surrounding the experimental evaluation.

many thanks   (area chair)

---

### Meta-Review · Area_Chair1 · 2018-12-15
**borderline, but lean in favor**

**Confidence:** 3
**Recommendation:** Accept (Poster)

**Metareview:**

Lean in favor

Strengths:  The paper tackles the difficult problem of automatic robot design. The approach uses graph neural
networks to parameterize the control policies, which allows for weight sharing / transfer to new policies even
as the topology changes.  Understanding how to efficiently explore through non-differentiable changes to the body
is an important problem (AC). The authors will release the code and environments, which will be useful in an area where there are
currently no good baselines (AC).

Weaknesses: There are concerns (particularly R2, R1) over the lack of a strong baseline, and with the results
being demonstrated on a limited number of environments (R1)  (fish, 2D walker). In response, the authors clarified the nomenclature and
description of a number of the baselines, and added others. AC: there is no submitted video (searches for "video" on the PDF text
produces no hits); this is seen by the AC as being a real limitation from the perspective of evaluation.
AC agrees with some of the reviewer remarks that some of the original stated claims are too strong.
  AC: the simplified fluid model of Mujoco (http://mujoco.org/book/computation.html#gePassive) is
unable to model the fluid state, in particular the induced fluid vortices that are responsible for a
good portion of fish locomotion, i.e., "Passive and active flow control by swimming fishes and
mammals" and other papers. Acknowledging this kind of limitation will make the paper stronger, not weaker;
the ML community can learn from much existing work at the interface of biology and fluid mechancis.

There remain points of contention, i.e., the sufficiency of the baselines. However, the reviewers R2 and R3 have
not responded to the detailed replies from the authors, including additional baselines (totaling 5 at present)
and pointing out that baselines such as CMA-ES (R2) in a continuous space and therefore do not translate in any obvious way
to the given problem at hand.

On balance, with the additional baselines and related clarifications, the AC feels that this paper makes a
useful and valid contribution to the field, and will help establish a benchmark in an important area.
The authors are strongly encouraged to further state caveats and limitations, and to emphasize why some
candidate baseline methods are not readily applicable.